# Breaking the Concentration Limit in Fluorescence Fluctuation Spectroscopy with Camera-Based Detection

**DOI:** 10.3390/ijms23179840

**Published:** 2022-08-30

**Authors:** Yu-Kai Huang, Per Niklas Hedde

**Affiliations:** 1Laboratory for Fluorescence Dynamics, University of California Irvine, Irvine, CA 92697, USA; 2Beckman Laser Institute and Medical Clinic, University of California Irvine, Irvine, CA 92697, USA; 3Department of Pharmaceutical Sciences, University of California Irvine, Irvine, CA 92697, USA

**Keywords:** fluorescence fluctuation spectroscopy, light sheet microscopy, molecular interactions, image correlation spectroscopy

## Abstract

Fluorescence correlation spectroscopy (FCS) is an extremely versatile tool that has been widely used to measure chemical reaction rates, protein binding, nanoparticle-protein interactions, and biomolecular dynamics in vitro and in vivo. As an inherently micro-sized approach, FCS is compatible with high-throughput screening applications, as demanded for drug design, but typically limited to nanomolar concentrations, which restricts possible applications. Here, we show how massively parallel camera-based detection with side illumination can extend the usable concentration range of FCS more than 100-fold to measure low affinity processes. Our line illumination (LIM) approach is robust, fast (1 s acquisition times), and does not require any reference measurements to characterize the observation volume size.

## 1. Introduction

By analyzing fluorescence fluctuations caused by molecules moving in and out of a tiny observation volume (~1 femtoliter), fluorescence correlation spectroscopy (FCS) can measure diffusion and reaction kinetics, the hydrodynamic radius, and the concentration of particles at the microscopic level [1]. Hence, FCS is an extremely versatile tool that has been widely applied to study chemical reaction rates, protein binding, nanoparticle-protein interactions, and biomolecular dynamics in vitro and in vivo [2]. As an inherently microscopic method, the minimal required sample volumes make FCS ideally suited for high-throughput screening applications, for example, in drug design [3]. While extremely sensitive at low concentrations (picomolar to nanomolar), FCS has been considered not applicable beyond the nanomolar concentration range, somewhat limiting its application. Recently developed nanofabrication, super-resolution microscopy, and near-field probing methods can extend the usable concentration range of FCS [4,5,6,7]. However, the specialized hardware required restricts widespread application of these methods. Here, we used a camera in combination with perpendicular illumination instead of commonly employed single point detectors to massively parallelize the measurement and to compensate for noise sources that typically limit the application of FCS at high concentrations. We show that our easy to implement approach extends the usable concentration range of FCS more than 100-fold. This allows for the study of binding processes in the micromolar concentration range, which can be important, for example, in drug design.

At high concentrations, FCS is generally considered to be noise-limited, as the reduced amplitude of molecular fluctuations becomes insignificant relative to other sources of fluorescence signal variations such as laser and detector noise. However, provided that the measured molecular brightness, i.e., the number of photons detected per molecule per time interval, stays constant, the signal-to-noise ratio of an FCS measurement should theoretically be independent of the concentration [8]. Yet practically, at high fluorophore concentrations, the limited count rate of highly sensitive single point detectors such as photomultipliers (PMTs) and avalanche photodiodes (APDs) demand a reduction of excitation intensity to avoid detector saturation, which in turn reduces the number of fluorescence photons emitted per molecule per time interval. This limitation in the measured molecular brightness effectively reduces sensitivity of FCS at high concentrations.

Nanofabrication, super-resolution, and near-field techniques overcome this problem by further reducing the femtoliter-sized observation volume such that the average number of molecules observed remains low, even at higher concentrations. Unfortunately, these approaches demand special nanosized sample containers, specific fluorophores, and/or powerful but expensive pulsed lasers, with only a moderate increase in the usable concentration range. Others have minimized laser noise by compensation with data from a photodiode monitoring the laser output power and extended the detector saturation limit by splitting the fluorescence signal onto up to eight APDs [9], which again requires a substantial technical effort.

Instead, we took advantage of the large number of pixels available with modern sCMOS cameras for parallelized detection in >16 lines of >240 pixels each recorded at >10,000 Hz. This massively parallel detection allowed us to process an extremely high photon flux and, at the same time, compensate for non-molecular signal fluctuations using spatial averaging within lines. To generate an axially confined observation volume, we illuminated the sample from the side with a thin beam (~1–2 µm diameter) that was then imaged onto a sCMOS camera similar to the geometries used in light sheet microscopy [10]. We found that this line illumination (LIM) approach was able to accurately measure the diffusion coefficient of a small dye molecule (Atto 488) in aqueous solution at concentrations up to 30 µM. In a first application, we then characterized the low-affinity binding of fluorescein isothiocyanate (FITC) to bovine serum albumin (BSA). Finally, we show how the LIM method can be used to measure the motion of fluorescently labeled biomolecules in live cells.

## 2. Results

### 2.1. LIM Method Validation

By illuminating with a Gaussian beam (Figure 1a), only one or a few lines of the camera chip are relevant to capturing the fluorescence from this thin, tubular observation volume and, in such region of interest, the camera frame rate could be increased to >10,000 Hz (<100 µs pixel dwell time). Each pixel in each line of this space-time (*x, t*) data set (Figure 1b) was then correlated with all other pixels in the same line to generate a correlation carpet of space and time lag (ξ, τ) (Figure 1c), to which a Gaussian function was fitted for each line (Figure 1d). By plotting the width of those Gaussians as a function of time lag (Figure 1e), the mean square displacement of the molecules was obtained, similar to the image mean square displacement (iMSD) analysis principle [11,12]. Notably, the only two parameters needed to quantify these data are the pixel size at the sample given by the objective effective magnification and the lag time between frames given by the camera frame rate. Contrary to single point FCS, no calibration of the observation volume size is needed to obtain the absolute value of the diffusion coefficient. We first validated our approach by measuring the (known) diffusion coefficient of a small dye in aqueous solution. For this purpose, Atto 488 was chosen because of its high water solubility, good photophysical parameters (QY = 0.8, ε = 90,000 in PBS at pH 7.4), and widespread use in fluorescence sensing applications. Like many small dyes of similar molecular weight (~0.8 kDa), Atto 488 diffuses at ~400 µm^2^/s in aqueous solution at room temperature [13], demanding a very high temporal resolution. The dye was diluted in PBS buffer at different concentrations in a range of 30 nM to 50 µM. Ten sets of 10,000 frames (1 s acquisition time for each set) were taken per concentration. The measured diffusion coefficients are shown in Figure 2a. The insets show photographs of the sample chamber filled with Atto488 solution was excited with 488-nm light. At a 30 nM dye concentration, the excitation beam/light sheet can be seen as faint green fluorescence trace within the solution. The cyan background visible in the photograph is a result of scattered 488-nm excitation light. At 50 µM of dye concentration, a >1000-fold increase, all excitation light is completely absorbed, and the resulting green fluorescence is much stronger. Figure 2b shows three example concentrations measured with the LIM method and corresponding single point FCS data fitted with a free diffusion model. Residuals are shown in Figure 2c,d. While the low correlation amplitude of single point FCS at a concentration >100 nM resulted in overestimated diffusion coefficients (510 ± 25 µm^2^/s at 300 nM), the MSD diffusion coefficient remained at the expected value up to 30 µM, a more than 100-fold increase of the measurement range. At higher dye concentrations, the signal-to-noise ratio of the measured correlation functions significantly declined, resulting in an overestimation of the diffusion coefficient at 50 µM (Figure 2a). We note that, at 50 µM, absorption of the excitation light by the highly concentrated dye (filter effect) prohibited the correct formation of the illumination beam at the focal plane of the detection lens. This effect is shown in the right inset of Figure 2a, where the excitation beam coming from the top right generated the strongest fluorescence immediately after entering the sample chamber and quickly declined towards the middle of the sample, where the signal was measured. In contrast, the fluorescence generated at 30 nM (left inset), was uniform across the entire illumination path. Thus, with an optimized sample geometry that reduces the relatively long path of the excitation beam through the sample solution before reaching the observation area, it may be possible to further increase the concentration range.

### 2.2. Binding at High Concentration

Next, we characterized the weak binding of fluorescein to bovine serum albumin (BSA) [14]. Fluorescein derivatives are widely used for protein labeling, while serum albumin (SA) is the most abundant soluble protein found in blood plasma. BSA is often used in biological investigations as a substitute for human serum albumin (HSA) and can interact with many biomolecules including amino acids, fatty acids, and many other small molecules, including drugs. Three structurally similar binding domains (I, II, and III) have been identified by crystallography [15]. To investigate the interaction of fluorescein and BSA at high concentrations, serial dilutions of BSA ranging from 0.10 μM to 1 mM were prepared, each containing 4 μM of fluorescein. The resulting diffusion coefficients measured by LIM at room temperature are shown in Figure 3. Over the measured concentration range, a reduction of the fluorescein diffusion coefficient from an initial 437 µm^2^/s (no BSA) to 135 µm^2^/s (1 mM BSA) was observed. Fitting of the data with a simple binding model resulted in a K_D_ of 178 ± 93 μM (mean ± SD). From the binding curve, the diffusion coefficient of the fluorescein-BSA complexes (molecular weight 67 kDa) was estimated at 68 µm^2^/s, a 6.4-fold reduction in the diffusion compared to free fluorescein (0.33 kDa) and a very reasonable result. For comparison, enhanced green fluorescence protein (EGFP), with 40% of the molecular weight (or 74% of the hydrodynamic radius) of BSA, was found to diffuse at ~90 µm2/s [11]. In principle, for fluorescein-BSA complexes, one would expect a ~5.9-fold reduction for a BSA monomer-fluorescein complex and a ~7.4 reduction for a BSA dimer-fluorescein complex with reference to free fluorescein. Given the millimolar BSA concentration needed for complete binding, one would expect most complexes to consist of BSA dimers as the monomer-dimer equilibrium of BSA has been measured with a dissociation constant of 10 ± 2 μM at 25 °C [16].

### 2.3. Application to Cells

Finally, we applied LIM to MCF10A cells over-expressing Arc-EGFP. Arc (Activity-regulated cytoskeleton-associated protein) or Arg 3.1 (Activity-regulated gene 3.1) is a key regulator of synaptic plasticity and required for long-term memory formation. In addition to a cytoplasmic fraction, Arc also accumulates in the nucleus where it is known to function as a transcriptional regulator. Using raster image correlation spectroscopy (RICS), we recently found that nuclear Arc displayed higher diffusivity than cytoplasmic Arc. As single point FCS, RICS is limited to nanomolar concentrations unless combined with super-resolution methods [17]. To validate application of the LIM method to live cells, we imaged MCF10A cells expressing Arc-EGFP in various locations within the cell. The cell nucleus was co-stained with NucBlue (Figure 4a) to be able to differentiate between nuclear and cytoplasmic Arc-EGFP (Figure 4b, boxes); an example LIM plot is shown in Figure 4c. We found an average diffusion coefficient of 3.0 µm^2^/s for the cytoplasm compared to significantly faster diffusion at an average of 8.8 µm^2^/s in the nucleus (Figure 4d), confirming the trend that we had previously obtained by means of RICS [18]. In addition to determining the particle dynamics, correlation methods can also measure the average concentration by means of the correlation function amplitude and the size of the observation volume. However, as with RICS, subtraction of the immobile fraction modifies the correlation function amplitude and absolute concentrations can no longer be measured accurately. However, relative concentrations can still be determined, and we found that, in the nucleus, the average Arc-EGFP concentration was ~6-fold higher than in the cell cytoplasm. Notably, our LIM measurements were carried out in cells 3D cultured in a collagen matrix as opposed to conventionally 2D plated cells that were used for the RICS experiments.

## 3. Discussion

The main reason why LIM can measure highly concentrated samples is because of the high saturation capacity of a sCMOS sensor compared to a PMT. The maximum tolerable photon count rates of PMTs are typically on the order of 10 MHz. This means that, for high fluorophore concentrations, the excitation light intensity has to be reduced to avoid saturating the PMT, which in turn results in a lower photon flux from each individual molecule observed. Typical CMOS camera pixels have a well depth of 12 Bit with 11 Bit usable analog to digital conversion (ADC) with a conversion factor of ~0.5 electrons per count, meaning that ~1000 values can be distinguished in a single readout cycle. Thus, at a ~10,000 Hz readout rate, which can be achieved when reading only a few lines of the sensor, a single camera pixel can match the PMT count rate of about 10 MHz. Therefore, by simultaneously exposing and reading >100 pixels in a single line, we can take advantage of much higher excitation intensities at high fluorophore concentrations without saturating the detector. In contrast to a PMT detector, the appropriate use of a CMOS camera sensor allows the maintenance of a high photon yield per molecule at high concentrations, which in the end critically determines the signal-to-noise ratio of the resulting correlation function.

## 4. Methods and Materials

### 4.1. Line Illumination

Our setup was based on the sideSPIM as previously described [10]. The output of a 488-nm laser (Omicron-Laserage, Rodgau-Dudenhofen, Germany) was spatially cleaned with a single mode fiber, reflected on a 2-axis galvo mirror system (Cambridge Technology, Bedford, MA, USA), and passed through a scanning lens (f = 50 mm) and tube lens (f = 180 mm) in a 4f configuration. A 4×, NA 0.16 objective lens (Olympus, Center Valley, PA, USA) was used the generate a Gaussian illumination beam with a waist size of 1.9 µm and a confocal parameter of 48 µm inside our custom sample chamber. Sample fluorescence was collected with a 60×, NA 1.0 water immersion lens (Olympus), separated from scattered excitation light with a 535/40 nm bandpass filter (Chroma, Bellows Falls, VT, USA), and focused via the internal tube lens of an inverted microscope body (IX71, Olympus) onto the chip of a sCMOS camera (PCO, Kelheim, Germany). The pixel size at the sample was 108 nm.

### 4.2. Single Point FCS

Single point FCS was measured with a Zeiss LSM880 equipped with the FCS module (Carl Zeiss, Jena, Germany). Sample fluorescence was excited with a 488-nm Argon laser line reflected off the internal dichroic (MBS488) and passed through a 40×, NA 1.2 water immersion objective (Zeiss). Fluorescence was focused onto a pinhole of 35 µm diameter (1 Airy unit) before detection within a range of 500 nm–600 nm. To calibrate the size of the point spread function, a 10 nM solution of Rhodamine 110 was prepared in PBS buffer and the beam waist of the resulting data was fitted by fixing the diffusion coefficient at 440 µm^2^/s [19]. For fitting of the correlation function, *G(τ)*, as a function of the lag time, *τ*, we used a free diffusion model
(1)$$G\left(\tau \right)=\frac{\gamma }{N}\;\frac{1}{1+\frac{4D\tau }{\omega^{2}}}\;\frac{1}{\sqrt{1+\frac{4D\tau }{S\omega^{2}}}}$$
with *N* the resulting average number of particles inside the observation volume, *γ* = 0.35 the correction factor for the Gaussian shape of the point spread function, *S* = 5 the aspect ratio (latera to axial extension), *ω* = 0.20 µm the lateral beam waist, and *D* the resulting diffusion coefficient, which is related to the diffusion time *τ_D_* = *ω*^2^/4*D*.

### 4.3. Dye and Protein Solutions

Atto 488 (#41051-1MG-F, Sigma-Aldrich, St. Louis, MO, USA) was dissolved in PBS buffer to a stock concentration of 100 µM, determined by measuring the absorption at 490 nm (Nanodrop, ThermoFisher, Waltham, MA, USA). Serial dilutions in PBS buffer of 30, 100, and 300 nM, as well as 1, 3, 10, 30 and 50 µM were placed in our custom imaging chamber. FITC (#1245460250, Sigma-Aldrich) and BSA (#A7906-10G, Sigma-Aldrich) were diluted in PBS buffer to generate stock solutions of 74 µM and 3 mM, as determined by absorption measurements at 490 nm and 280 nm, respectively. For both solutions, the pH was adjusted to 9 by adding 5 N NaOH to increase the fluorescence quantum yield of FITC.

### 4.4. Cell Sample Preparation

MCF10A (American Type Culture Collection, Manassas, VA, USA) cells were cultured in DMEM/F12 with high glucose, sodium pyruvate, and L-glutamine (Thermo Fisher Scientific, Waltham, MA, USA) supplemented with 5% horse serum (Thermo Fisher Scientific), 20 ng/mL epidermal growth factor, 0.5 mg/mL Hydrocortisone (Sigma-Aldrich), 100 ng/mL cholera toxin (Sigma-Aldrich), 10 μg/mL insulin (Sigma-Aldrich), and 1% Penicillin-Streptomycin 100× solution (Genesee Scientific, San Diego, CA, USA). Cells were plated in 35 mm dishes and transfected with plasmid encoding Arc-EGFP using Lipofectamine 3000 according to the manufacturer’s protocol (Thermo Fisher Scientific). Collagen type I (Corning, Corning, NY, USA) gels were prepared at 2 mg/mL concentration by dilution with cell culture medium at 7.2 pH followed by polymerization inside sideSPIM imaging chambers [10] for 1 h at 37 °C. Cells were subsequently transferred to collagen type I gels and incubated for 12 h before imaging to allow for cell attachment to the gel. To visualize the cell nuclei, DNA staining with NucBlue (#R37605, ThermoFisher) was performed 15 min before imaging.

### 4.5. Measurement Parameters

Single point FCS data were measured at 1.7 mW laser power with 10 s acquisition times. For LIM analysis of Atto 488 and FITC binding to BSA, the laser output at the objective lens was set to 6 mW and 16 lines of 260 pixels were acquired at 10,246 Hz. A total of 100,000 frames were imaged for each data set (10 s acquisition time). For LIM analysis of Arc-EGFP in MCF10A cells, the laser power was reduced to 0.2 mW to prevent photobleaching. Due to the slower diffusion of Arc-EGFP in cells compared to the diffusion of free dye in aqueous solution, the camera exposure time was increased to 2 ms (500 Hz) and a total of 5000 frames were acquired for each cell.

### 4.6. Data Analysis

Each data set, $I\left(x,y,t\right)$, was reordered, $I\left(x,t,y\right),$ with ImageJ 1.53c and loaded into Matlab R2019a (Mathworks, Natick, MA, USA) for analysis, with 100,000 frames (*t*) of 16 lines (*y*) with 260 pixels (*x*). For each line, the spatial averages, ${\langle I\left(x,t,y\right)\rangle }_{x}$, and the spatio-temporal averages, ${\langle I\left(x,t,y\right)\rangle }_{x,t}$, were calculated and used to compensate for intensity fluctuations not attributed to molecule movement, such as laser fluctuations. The spatial averages represent the average intensity of all pixels in each line at any given time point and are therefore an intrinsic measure of the laser power. By dividing by the spatial averages and multiplying with the spatio-temporal averages, i.e., the time average of the average intensity of all pixels in each line, the data can be normalized for the laser power:(2)$$J\left(x,t,y\right)=\frac{I\left(x,t,y\right)}{{\langle I\left(x,t,y\right)\rangle }_{x}}\cdot {\langle I\left(x,t,y\right)\rangle }_{x,t}$$

Next, the immobile fraction (if present) was removed by subtracting the temporal average in each line. To avoid negative values in the resulting data set, the global average was subsequently added:(3)$$F\left(x,t,y\right)=J\left(x,t,y\right)-{\langle J\left(x,t,y\right)\rangle }_{t}+{\langle J\left(x,t,y\right)\rangle }_{x,t,y}$$

For each line of the 16 lines × 260 pixels × 100,000 frame data set, *F_y_(x,t)*, these data were then space-time correlated:(4)$$G_{y}\left(\xi ,\tau \right)=\frac{<F_{y}\left(x,t\right)\cdot F_{y}\left(x+\xi ,t+\tau \right)>}{{\langle F_{y}\left(x,t\right)\rangle }^{2}}-1$$

To the resulting correlation function, a Gaussian model,
(5)$$H_{\tau }\left(\xi \right)=A\cdot \exp\left(-{\left(\frac{G_{\tau }\left(\xi \right)-\xi_{0}}{w}\right)}^{2}\right)+B,$$
was fitted for each lag time, τ, with amplitude, *A*, offset, *B*, and width, *w*. The square of the fitted width, $w^{2}$, which corresponds to the particle mean square displacement, was then plotted as a function of the lag time and the slope was determined by linear regression. From the slope, the diffusion coefficient was calculated as previously described [11].

## 5. Conclusions

Fluorescence fluctuation spectroscopy (FFS)-based methods are elegant concepts but rely on appropriate modeling of the underlying molecule dynamics, and often require careful calibration. In contrast, (3D) single particle tracking (SPT) represents a more straightforward approach to capture molecular motion but requires spatially well separated molecules. Depending on the application, one technique may be preferred over the other. For example, SPT can be advantageous in accurately tracing and locating binding sites such as DNA transcriptional factors. FFS, on the other hand, is a very powerful method to measure average molecule diffusion, which works better in relatively large-scale applications such as drug screening and diffusion mapping of live cells. Using camera-based detection, we were able to extend the usable concentration range of FFS ~100-fold without the need for super-resolution, near-field, or nanofabrication methods. As an added benefit, the LIM method does not require calibration of the observation volume to yield absolute diffusion coefficients and, due to the high photon flux, data points can be acquired in as little as 1 s. Using the LIM approach, we were able to establish the dissociation constant of the BSA-fluorescein binding equilibrium at 178 ± 93 μM of BSA.

## Figures and Tables

**Figure 1 ijms-23-09840-f001:**
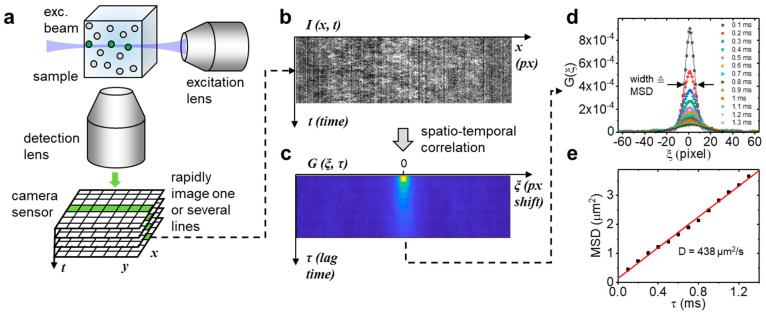
Line illumination principle. (**a**) A thin beam of light is injected into the sample from the side and fluorescence is collected perpendicular to the beam direction with a high numerical aperture lens. A rapid time series of the resulting signal is collected with a fast camera. (**b**,**c**). One or multiple lines are extracted from this data set followed by spatiotemporal correlation of the space-time series. (**d**,**e**) The correlation function for each lag time is fitted with a Gaussian distribution. The width of this distribution corresponds to the mean square displacement of the molecules under study and plotted as a function of the lag time, the absolute diffusion coefficient can be determined from the slope.

**Figure 2 ijms-23-09840-f002:**
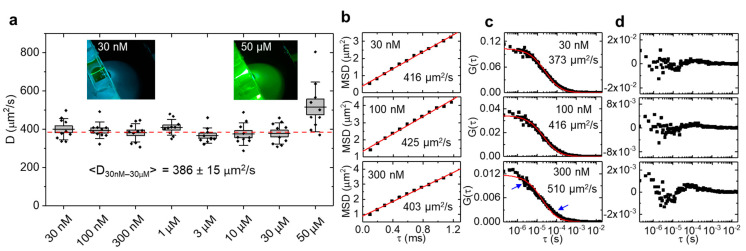
Diffusion of Atto 488 in aqueous solution measured by LIM. (**a**) For each concentration, 10 measurements of 1 s duration were acquired. Lines = averages, boxes = standard errors, whiskers = standard deviations. Dashed line = average diffusion coefficient (386 ± 15 µm^2^/s, mean ± SD) of all measurements excluding 50 µM. Insets are photographs of Atto 488 fluorescence at 30 nM and 50 µM. (**b**) Example mean square displacement (MSD) of 1 s LIM data. (**c**) Exemplary 10 s single point FCS data of the same solutions fitted with a free diffusion model, and (**d**) corresponding residual plots. At 300 nM, single point FCS overestimates the diffusion coefficient and deviates from the free diffusion model (see arrows).

**Figure 3 ijms-23-09840-f003:**
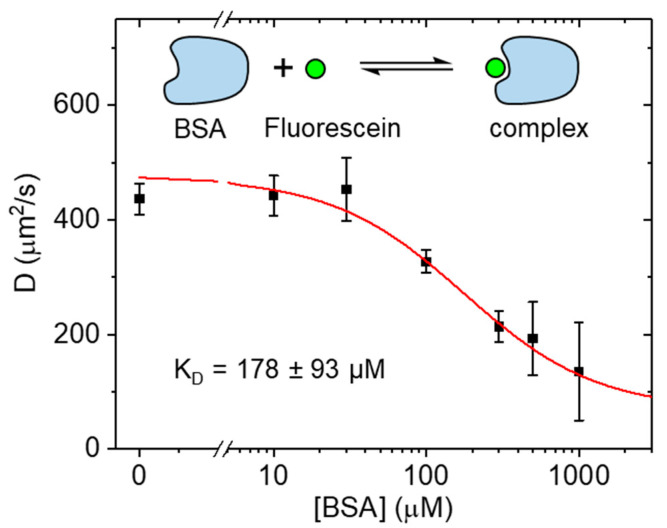
Diffusion coefficients of fluorescein mixed with BSA at different concentrations. When bound to BSA, the diffusion coefficient was much reduced compared to free fluorescein. With 4 µM of fluorescein, a K_D_ of 178 ± 93 µM BSA was obtained (mean ± SD).

**Figure 4 ijms-23-09840-f004:**
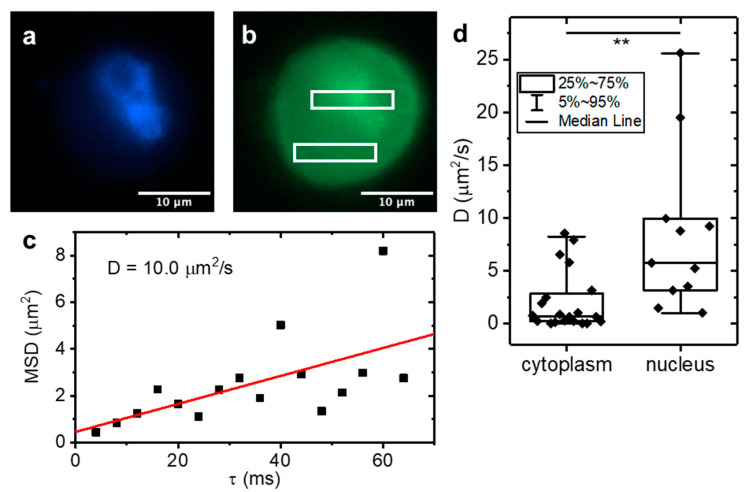
Diffusion coefficients of Arc-EGFP measured in MCF10A cells. LIM applied to 3D cultured MCF10A cells expressing Arc-EGFP. (**a**) Example fluorescence image of NucBlue nuclear stain and (**b**) Arc-EGFP of a single cell. (**c**) Example LIM data of Arc-EGFP diffusion. (**d**) Diffusion coefficients were measured in the nucleus and cytoplasm. Box plots show the median ± interquartile ranges. Significantly faster Arc-EGFP diffusion was found in the nucleus compared to the cell cytoplasm (** indicates *p* = 0.0075, unpaired two sample *t* test, N = 20).

## Data Availability

Data may be obtained from the authors upon reasonable request.

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
