# Peer review of "Breaking the Concentration Limit in Fluorescence Fluctuation Spectroscopy with Camera-Based Detection"

_ijms, 2022, doi:10.3390/ijms23179840_

Round 1
Reviewer 1 Report
These authors (Huang and Hedde) demonstrated an improved method of fluorescence fluctuation spectroscopy using light-sheet illumination. The manuscript is written well; in addition, the concept and its demonstration examples are very nice. However, to show and emphasize the advantage of the improved method clearly, I suggest some revisions.
In the result section, these authors describe the disadvantage and limitations of conventional single beam FCS; however, it is still unclear why the LIM can overcome the disadvantage. The principle of why LIM can be applied to measure highly concentrated samples should be clearly described.
In the legend of Figure 1, especially d and e, although a typical analysis procedure is presented, I judge that there is a lack of explanation regarding the mathematical relationships between the figures. I think one of the reasons why it is difficult to understand is that the original xi = zero is blindly defined in d, even though its location in Figure c is not explicitly indicated. In Fig. 1e, it would be better to specify the value of the diffusion coefficient obtained from the slope, even if it is just an example. What do you think?
In Figure 2a, it is very difficult to distinguish what is represented in the inset. In the left photo, the color seems to be cyan, but green in the right one. Are you using a pseudo-color? How can we observe these inset figures to convey the author's point of view? It is very ambiguous.
In Figure 2c, they showed an example of how the error is specifically observed; though I can distinguish the position, it is not general. How about showing the residual plot?
In Figure 2a, the reason why the diffusion coefficient is not accurately measured for the 50 uM ATTO488 sample is the same as for a single-point FCS. Or is there another reason? I recommend that you add not only successful results (advantages), but also steady consideration.
Figure 3 is a very nice demonstration. This is just a comment.
In Figure 4, how about estimating the concentration (e.g., expression levels in cells)? It is very nice information on how much concentration using LIM can be measured even in cells. Otherwise, you can use a conventional FCS.
Minor points:
Line 9, the bold 'fluorescence' should be corrected.
The entire conditions (e.g., structure parameter and diffusion time of the standard dye) when using LSM880-FCS should be described in the “measurement parameters” section of the method.
Reviewer 2 Report
This manuscript proposed fluorescence correlation spectroscopy by using the fast linearly scanning of sCMOS. The detecting concentration range is significantly improved. Overall, this manuscript is well written. I have some concerns and suggestions:
1. Please clarify the spatial averages and spatio-temporal averages. It’s still not clear to me how you calculate them.
2. On page 3 line 140, does the global average contains the immobile fraction?
3. Please define ?? (?,?).
4. Where is τ in equation (4)?
5. The unit of MSD should be µm2.
6. A brief comparison between FCS and recently emerged real-time 3D single molecule tracking methods would be appreciated since both aim to obtain the diffusive information of molecules.
Round 2
Reviewer 1 Report
Well done.
Reviewer 2 Report
The authors solved all my concerns. Therefore I support this manuscript's publication.